# Immunotherapy for Cervical Cancer: Are We Ready for Prime Time?

**DOI:** 10.3390/ijms23073559

**Published:** 2022-03-24

**Authors:** Margherita Turinetto, Anna A. Valsecchi, Valentina Tuninetti, Giulia Scotto, Fulvio Borella, Giorgio Valabrega

**Affiliations:** 1Department of Oncology, University of Turin, Ordine Mauriziano Hospital, 10128 Turin, Italy; dr.ssavalsecchianna@gmail.com (A.A.V.); dr.ssatuninettivalentina@gmail.com (V.T.); giulia.scotto@edu.unito.it (G.S.); giorgio.valabrega@unito.it (G.V.); 2Gynecology and Obstetrics 1, Department of Surgical Sciences, City of Health and Science, University of Turin, 10100 Turin, Italy; fulvio.borella87@gmail.com

**Keywords:** cervical cancer, immunotherapy, immune checkpoint inhibitors, therapeutic vaccines, TILs, tisotumab vedotin, human papillomavirus

## Abstract

The prognosis of invasive cervical cancer (CC) remains poor, with a treatment approach that has remained the same for several decades. Lately, a better understanding of the interactions between the disease and the host immune system has allowed researchers to focus on the employment of immune therapy in various clinical settings. The most advanced strategy is immune checkpoint inhibitors (ICIs) with numerous phase II and III trials recently concluded with very encouraging results, assessing single agent therapy, combinations with chemotherapy and radiotherapy. Apart from ICIs, several other compounds have gained the spotlight. Tumor Infiltrating Lymphocytes (TILs) due to their highly selective tumoricidal effect and manageable adverse effect profile have received the FDA’s Breakthrough Therapy designation in 2019. The antibody drug conjugate (ADC) Tisotumab-Vedotin has shown activity in metastatic CC relapsed after at least one line of chemotherapy, with a phase III trial currently actively enrolling patients. Moreover, the deeper understanding of the ever-changing immune landscape of CC carcinogenesis has resulted in the development of active therapeutic vaccines. This review highlights the different immunotherapeutic strategies being explored reflects on what role immunotherapy might have in therapeutic algorithms of CC and addresses the role of predictive biomarkers.

## 1. Introduction

Cervical cancer (CC) is the fourth most frequent cancer in women: it is diagnosed in approximately 570,000 women worldwide every year. Invasive CC is responsible for about 342,000 deaths annually [1] from 2007 to 2016, the death rate decreased by about 1% per year in women over 50 years old but was stable in women under 50 years old. The mortality is different across various countries, from less than 2 per 100,000 in Australia and New Zealand to more than 22 per 100,000 in some African countries [1]. Therefore, cervical neoplasm still represents a health concern worldwide, particularly in developing countries where screening programs with Papanicolaou (PAP) test and vaccination campaigns are not widely available [2].

The Human papillomavirus (HPV) is responsible for up to 15.4% of human cancers attributable to carcinogenic infections [3] and almost all cases of CC, with more than 70% being linked to high-risk HPV types 16 and 18 [4]. The key event to HPV induced carcinogenesis is the integration of the viral genome into a host chromosome.

The prognosis of diagnosed invasive CC remains poor, especially for late stages and relapsed disease; early-stage CC has a 5-year survival rate of 91%, while advanced or recurrent disease, not amenable to surgery or radiotherapy, has a 5-year survival rate of 17% [5], with median progression free survival (PFS) of 2–5 months and overall survival (OS) of 5–16 months [6]. Moreover, for the early stages the risk of disease recurrence is 10–20%, while for advanced stages the risk is about 50–70% [6]. In localized CC, surgery or a combination of chemotherapy and radiation represent the frontline treatment. According to the National Comprehensive Cancer Network (NCCN) guidelines, the choice of the best therapeutic approach depends on the stage of disease and clinicopathologic risk factors. International Federation of Gynecology and Obstetrics (FIGO) stages IA1 to IIA can be treated with radical hysterectomy, with lymphadenectomy being considered if lymphovascular space invasion is present or if IA2 and above. Fertility sparing surgery can be considered in IA1 women with a strong reproductive desire after appropriate counseling [7]. For locally advanced CC, cisplatin-based chemoradiotherapy (CRT) is considered the standard of care [8,9]; while for advanced and recurrent CC, the treatment of choice has been a platinum-based doublet +/− bevacizumab, up until the recent developments. In 2014, Tewari et al. demonstrated that in this setting of disease bevacizumab in combination with chemotherapy shows an advantage in OS of 4 months compared with chemotherapy alone (17 months versus 13 months) and a 12% higher response rate (48% versus 36%) [10]. In the case of disease progression, the treatment options for the second line are very limited and include single-agent therapies, such as taxanes, topotecan, gemcitabine, vinorelbine with a disappointing median response rate of only a few months, median OS of 9.3 months and median PFS of 3.2 months [11,12].

Considering the increasing evidence of immunotherapy in advanced CC, we aim to present a comprehensive and updated overview on the main immune-based strategies currently being explored and on what role immunotherapy might have in therapeutic algorithms (Figure 1).

## 2. The Rationale for Immunotherapy in Cervical Cancer

The rationale for employing immunotherapy in CC is strongly supported by multiple molecular features, such as high tumor mutational burden (TMB), microsatellite instability (MSI), high expression of programmed death ligand 1 (PD-L1) and high tumor inflammatory state.

It is now established that CC can be considered a tumor with a high TMB [13]. The number of somatic mutations per DNA megabase is considered a proxy for neoantigen burden, which is a newly established independent predictor of immune checkpoint inhibitor treatment outcome [14].

Furthermore, Lazo et al. reported that 8% of cervical cancers present MSI [15], an already established predictive marker for response to immunotherapy in other tumors, such as endometrial cancer [16].

In HPV-driven carcinogenesis, HPV positive cells are able to evade immunosurveillance via the inhibition of acute inflammation and immune recognition. Recent studies show that this viral and inflammatory cancer setting could be responsible for the induction of PD-L1 expression [17]. In fact, PD-L1 expression seems to have an important role in creating an “immune-privileged” site for initiation and persistence of HPV infection by downregulating T cell activity and generating an adaptive immune resistance [18,19]. In particular, high levels of PD-L1 expression are reported in 35–96% of cervical cancers [18,20]: its presence is rare in normal cervical tissue, but it is detectable in both cervical cancer T-cells and tumor cells.

As mentioned before, HPV genome integration is a vital step towards carcinogenesis, often occurring at more fragile sites of the human DNA. While there are not specific spots where integration happens more often, a more recognizable pattern is found for the viral genes; loss of the transcriptional repressor E2 might lead to the deregulation of E6 and E7, that are consistently maintained and have been extensively documented as oncogenic genes, inactivating p53 and pRB tumor suppressors. The expression of E6 and E7 genes is not only vital for the initial stages of the premalignant lesion but actively contributes to progression through the induction of genomic instability [21].

Very promising and interesting results have been obtained in various applications of immunotherapy; as of now, the class of compounds that has advanced the most in research is immune checkpoint inhibitors (ICI), with several phase II and III trials currently ongoing (Table 1).

## 3. Immune Checkpoint Inhibitors (ICI): Mechanism of Action

The usage of exogenous immune compounds, such as immune checkpoint inhibitors (ICI), with Cytotoxic T lymphocyte antigen 4 (CTLA4) and programmed cell death 1 (PD-1) being the most commonly targeted molecules, has so far been proven to be an effective option for a number of tumors (NSCLC, bladder cancer, melanoma) and is currently being explored in CC.

T cells are activated via a series of parallel mechanisms; among those, the binding of CD28 to B7 on APC; CTLA-4 is a homolog of CD28 but carries a higher affinity to B7. When CD28-B7 is relatively less represented than CTLA-4-B7, activation will not be achieved because of the reduction of IL-2 production and because of inhibitory signals counteracting TCR-MHC binding [22]. CTLA-4 regulates effector T cells through other pathways as well, such as its constitutional expression on Tregs, that downregulate B7 expression on APCs.

PD-1 is a costimulatory receptor typical of exhausted T cells, expressed by the cell in situations of prolonged stimulation, such as chronic inflammation or cancer; its binding to PD-L1 decreases the production of IFN-γ, TNF-α, IL-2 and prevents the propagation of the TCR signaling. PD-1 expression on Tregs has been shown to stimulate CD4+ T cell conversion into the regulatory phenotype [23]. The location of the two molecules differs; PD-1 is present on T cells, B cells and myeloid cells, while CTLA-4 is limited to T lymphocytes. Ligands of PD-1 are also widely expressed, with PD-L1 and 2 being found on non-hematopoietic cells and non-lymphoid tissues as it is induced by pro-inflammatory and tumorigenic signals [24]. Although our understanding of the molecular mechanisms behind the efficacy or failure of ICI therapy is still developing, it is safe to say that a successful antitumoral response requires the processing and presentation of tumoral antigens by activated APCs or dendritic cells; these MHC-bound antigens are then recognized by unique T cell receptors, providing the first signal for T cell activation, that will be implemented by costimulatory signals, such as CD28-B7. A subset of CD8+ T cells will differentiate into effector T cells, actively killing cells displaying the tumoral antigen. For long-term immunity, effector T cells must also differentiate into memory T cells, dependent on the interaction with CD4+ helper T cells and dendritic cells [25].

Resistance to ICI can, therefore, be a consequence of a defect of any of the steps mentioned.

A high tumor mutational burden is widely recognized as a predictive marker for response to ICIs, because of the enhanced production of neoantigens caused by genetic instability; several tumor intrinsic or extrinsic mechanisms can negatively influence the production or the recognition of neoantigens. Genetic or epigenetic alterations can lead to the loss of expression of the neoantigens, and disruption of the antigen processing and presenting apparatus, such as the loss of B2M or HLA class I molecules, which can account for primary resistance to ICIs even in tumors with high antigenicity [26].

Insufficient T cell generation can also be connected to low intra-tumoral immune infiltration; the loss of PTEN [27] increases expression of CCL2 and VEGF, beta catenin-WNT signal alterations upregulate CCL4 [28], loss of STK11/LKB1 in KRAS mutated cells promotes IL-6 production [29].

Interestingly, specific gene signatures have been identified in non-responding patients, such as IPRES (innate PD-1 resistance) comprised of proangiogenic factors, pro EMT transition factors, immune-suppressive cytokines. Of note, genes connected to resistance to MAPK inhibitors and BRAF-MEKi inhibitors characterize IPRES [30].

Association of ICIs to other strategies, such as chemotherapy and radiation to promote cell death, and therefore, antigen presentation or target therapies that block immunosuppressive pathways (e.g., VEGF, NGF-beta) are a promising approach to overcoming resistance due to insufficient neoantigen presentation [31].

After successful T cell priming, an unfavorable TME may impair function; high levels of immune suppressor cytokines, recruitment of immunosuppressive and regulatory cells and expression of inhibitory receptors (PD-1, CTLA-4) may effectively impact cytotoxicity. Loss of JAK-STAT pathway components was detected in patients with secondary resistance to ICI [26], while IFN-gamma receptor deletion resulted in anti PD-1 resistance in murine models [32]. Preclinical data does suggest a more complex role of interferon signaling in determining resistance or response to ICI. Functional exhaustion of T cells is a well-documented component of immune escapism; however, different subsets of exhausted CD8+ T cells respond differently to ICI. A specific subpopulation of CTLA4+ PD1+ CD8+ partially exhausted T cells expand in response to PD1-PDL1 blockade and are characterized by features shared by follicular T cells and stem cells [33]. While anti PD1-PDL1 therapy stimulates CD8+ T cells, anti CTLA-4 reinvigorates Th1, such as CD4+ cells [34]. Therefore, further studies are needed to better target specific subsets of T cell populations, suggesting a strong preclinical rationale for combination therapy.

## 4. ICI: Clinical Development

The use of ICI in CC has been a topic of interest in the last few years, with many trials employing immunotherapy in all clinical settings of the disease: localized, advanced and recurrent.

The only ICI currently approved by the Food and Drug Administration (FDA) is Pembrolizumab: on 12 June 2018 approval was granted for patients with recurrent or metastatic PD-L1 positive cervical cancer with disease progression during or after chemotherapy. Pembrolizumab is a highly selective, fully humanized monoclonal antibody that blocks the PD-1 receptor expressed on T cells and inhibits the PDL1 pathway [35]. The study that led to the approval was the phase II basket trial Keynote-158 [36]. Enrolled participants had unresectable or metastatic solid tumors that failed to respond to standard of care therapy. The ORR was 12.2% (95% CI, 6.5 to 20.4%) with 10.2 months of median follow-up and increased to 27% for patients with longer follow-up (at least 27 weeks). In particular, three patients achieved complete response (CR) and nine patients achieved a partial response (PR): all these 12 responses were in patients with PD-L1-positive tumors with an ORR of 14.6% (95% CI, 7.8 to 24.2%). Grade 3 to 4 AEs were reported in 12.2% of patients and the most common ones have increased alanine aminotransferase and aspartate aminotransferase; no deaths were reported.

As for trials in the setting of localized CC, curable with CT-RT concomitant therapy, the CALLA phase III trial has recently completed enrolment of 770 patients, who were randomized to either CT-RT or CT-RT plus durvalumab.

Additionally, worth mentioning and still actively enrolling patients, Keynote A18 aims to treat 980 patients affected by locally advanced chemo naïve CC with CT-RT and pembrolizumab or placebo.

As for stage IVB or recurrent CC, a clinical setting where the standard of cure is comprised of platinum, paclitaxel, or topotecan, with or without bevacizumab, Keynote 826′s first interim analyses results were very recently presented at ESMO 2021. The addition of pembrolizumab to 1st line CT, with or without bevacizumab, led to a 33% reduction in the risk of death and a 35% reduction in the risk of disease progression and death. The advantage in PFS and OS is present regardless of PD-L1 status and the use of bevacizumab. Adverse events were manageable, with grade 3 or higher events in 81.8% of the pembrolizumab arm vs. 75.1% in the placebo arm. The most common adverse events of grade 3 or higher were anemia (30.6% with pembrolizumab vs. 26.9% with placebo) and neutropenia (12.4% vs. 9.7%).

These very relevant numbers certainly make Keynote 826 a milestone study at this point in time.

Patients affected by CC who relapse or do not respond to 1st line CT can currently be treated with monotherapy CT (gemcitabine, pemetrexed, vinorelbine…), with a median PFS of only a few months. Currently active in this setting is the BRAVA trial, comparing balistilimab with a physician’s choice of CT, and the EMPOWER trial, featuring cemiplimab vs. CT.

BRAVA has recently completed enrolment. The results from the phase I–II (NCT03495882) trial, show that balistilimab was well tolerated and active, with an ORR of 14% and a median DOR of 15.4 months are worthy of further research on a wider sample, which BRAVA will provide.

Agenus seemed on its way to FDA approval but the company is now pulling the application after the agency granted full approval to Merck’s pembrolizumab (Keytruda) in the same indication.

The EMPOWER’s secondary interim analyses were presented at ESMO 2021, and the efficacy demonstrated by cemiplimab allowed the trial to be terminated early; OS was of 12 months vs. 8.5 in the overall population, with a slight difference between histotypes (11.1 vs. 8.8 in the squamous carcinoma and 13.3 vs. 8.7 in the adenocarcinoma population).

### 4.1. Combination of PD-L1 Inhibition and CTLA-4 Inhibition

Different studies have investigated the role of combinatorial approaches that block both PD-1 and CTLA-4 pathways in cervical cancer with promising results. This paragraph aims to present an overview of the main trials, concluded and ongoing, about this possible future therapeutic strategy.

First, the CheckMate 358 study investigated the efficacy that the combination of nivolumab (anti-PD-1) and ipilimumab (anti-CTLA-4) can have in women with squamous cell carcinoma (SCC) of the cervix who had received no more than two prior systemic therapies for advanced disease [36,37].

The ICI association has been explored in two different regimens: the first one was nivolumab 3 mg/kg every 2 weeks plus ipilimumab 1 mg/kg every 6 weeks (Combo A); while the second one was nivolumab 1 mg/kg plus ipilimumab 3 mg/kg, given every 3 weeks for four doses followed by nivolumab 240 mg every 2 weeks (Combo B), for ≤24 months. On one hand, for women treated with the first regimen, ORR was 31.6% for those with no prior treatment and 23.1% for patients with previous treatment. On the other hand, for women treated with the second combination regimen, the ORR was 45.8% for those with no prior treatment and 36.4% for those with prior treatment. So, the results were promising especially in patients who had not received previous treatments. The efficacy and the durable clinical activity of these two dosing schedules were found regardless of the expression of PD-L1. In addition, the ICI association demonstrated manageable toxicity and safety profiles: the incidence of grade 3–4 TRAEs was 28.9% in Combo A and 37.0% in Combo B, respectively [38].

A phase I trial evaluated the safety and tolerability of durvalumab (anti-PD-L1) and tremelimumab (anti-CTLA-4) in subjects with advanced solid tumors, including CC, who progressed after standard treatment; 46.2% of patients experienced SD and the most frequent grade ≥ 3 TRAEs were represented by diarrhea and colitis [39].

There also is an ongoing phase II clinical trial which is investigating balistilimab (anti-PD-1) in association with zalifrelimab (anti-CTLA-4) (NCT03495882, See Table 1) in patients with metastatic or locally advanced CC who already received first-line platinum-based chemotherapy. ORR of the single-agent balistilimab was 14% with three complete responses (CRs) and 20 partial responses (PRs) in NCT03495882. The ICI combination showed an increased ORR of 22%, with eight CRs and 23 PRs. The median DOR was 15.4 months with balistilimab monotherapy, while the mean DOR was not reached in the combination regimen. The responses were more frequent in patients with PD-L1-positive tumors and in squamous cell histology. Both single-agent balistilimab and balistilimab-zalifrelimab combination therapy were well tolerated with about 30% and 35% of subjects with an AE, respectively. Treatment discontinuation was 13.7% in monotherapy and 10% in the combination regimen. Two deaths occurred, both in the combination trial because of nephritis and pneumonitis [40].

### 4.2. Combination of Immunotherapy and Antiangiogenic Agents

Anti-vascular endothelial growth factor (VEGF) has emerged as a therapeutic target in several malignancies, including cervical cancer where it’s considered a growth factor responsible for the proliferation, migration and survival of endothelial cells [41]. In fact, high levels of VEGF have been associated with advanced stages of cervical cancer [42]. Hence, combinatorial approaches that target both PD-1 and VEGF have been evaluated in recent clinical trials. Zhang et al. demonstrated that the combination of immunotherapy and antiangiogenic agents is able to guarantee at least an additive effect in mouse models [43].

BEATcc, an ongoing trial that is exploring the addition of atezolizumab to platinum chemotherapy, paclitaxel and bevacizumab in 404 patients with Stage IVB, persistent or recurrent CC [44], has recently completed its enrolling phase and results are anxiously awaited.

A phase II study is exploring the association of camrelizumab (anti PD-1) and apatinib (tyrosine kinase inhibitor of VEGF receptor 2) in patients with advanced CC who progressed after at least one line of systemic therapy. The combination seems promising: ORR, the primary endpoint, was 55.6% with two complete responses and 23 partial ones; median duration of response and median OS was not reached and median PFS was 8.8 months. Toxicities were noticeable but manageable: about 71.1% of patients had grade ≥ 3 TRAE and the most frequent adverse events were hypertension (24.4%), anemia (20.0%) and fatigue (15.6%) [45].

Another ongoing phase III randomized double-blind study is the FERMATA trial. It is currently evaluating the efficacy and safety of BCD-100 (Anti-PD-1) in combination with platinum-based chemotherapy with and without bevacizumab as a first-line treatment of subjects with advanced CC (NCT03912415).

Although most results are still pending, concurrent blockage of different pathways could offer new promising first or second line treatment opportunities in CC, especially considering the manageable safety profile.

### 4.3. Combination of Immunotherapy and Radiation: The Abscopal Effect

Radiation therapy can theoretically prime the immune system by releasing tumor neoantigens, thus creating an adaptive and cytotoxic T-cells mediated immune response that acts locally and distally.

The ability of radiation therapy to enhance, through immune priming, local tumor control and systemic response is known as the abscopal effect; however, ionizing radiation also induces immunosuppressive signals, mobilizing Tregs, M2 macrophages and myeloid-derived suppressor cells, dampening the long-term effect on the immune system. Stemming from this concept, the combination of RT and immunotherapy, such as ICI could prevent immunosuppressive signals and enhance the abscopal effect [46].

Strong evidence of its effectiveness in clinical settings is still needed, with promising results in NSCLC, melanoma and renal carcinoma [47], hence why many clinical trials are underway in order to investigate which is the optimal radiation timing, dose and technique to combine with immunotherapy in order to induce immunogenic tumor cell death, both for unresected local disease and distant micrometastatic spread [48].

## 5. Tumor Infiltrating Lymphocytes (TILs)

Implementation of Tumor infiltrating lymphocytes (TILs; Figure 2.) has proven to be one of the most effective and specific adaptive cell therapies (ACT). Very promising results have been obtained in various settings (melanoma, breast cancer), with less clinical data available for gynecologic malignancies, such as CC [49].

On the other hand, the prognostic and predictive implications of the characteristic of a patient’s TILs in CC have been more specifically addressed.

Various populations are present, such as CD4+ and CD8+ T cells, γδ T cells, B cells, natural killers, and Treg cells. Although TILs activity is often impaired by the immunosuppressive TME, a higher prevalence of CD4+ and CD8+ correlates with a better outcome, and a thorough definition is important as it is becoming a promising predictive marker to better select ICI responders.

Immune cell populations that are tumor specific have a much stronger tumor killing ability when compared to immune cells from blood, therefore, through the removal of potentially active T cells, such as CD4+ and CD8+ from an immunosuppressive environment and their expansion in a permissive in vitro environment, the aim of TILs therapy is to restore anti-tumor immunity. CD8+ TILs are able to kill tumor cells both directly when presented with neoantigens and indirectly by activating apoptosis-inducing FAS-FASL pathways [50], and differentiate into tissue resident memory T cells when stimulated by IL-15 and TGF-β [51]. CD4+ T cells can differentiate into different phenotypes, such as Th1, Th2, Th17 and Treg, playing a regulatory role; the Th1 phenotype stimulates CD8+ T cells cytotoxic activity through the production of IFN-γ.

Studies have observed a correlation between the higher presence of CD8+ T cells and response to treatment. Martins et al. showed a more intense tumor infiltrating population in responders to chemoradiation compared to non-responders [52]. A small pilot study compared TILs present in the tumor stroma after neoadjuvant CT with either cisplatin-paclitaxel or cisplatin only, noticing an increase in CD8+ cytotoxic T cells and a decrease in Ki67+CD3+CD8− T cells only in the cases treated with combination therapy [53].

RT might also have an impact on TILs and therefore be immune activating; both patients undergoing CT-RT treatment [54] and locally advanced cases undergoing EBRT and brachytherapy with or without CT [55] showed an increased CD4+ and CD8+ population in TILs and a correlation with a better OS (5-year OS: 53.8% versus 23.8%, *p* = 0.038).

So far very few clinical studies have been conducted utilizing TILs therapy in CC. Stevanović et al. [56] in 2015 treated nine patients with metastatic CC, progressed after platinum-based chemotherapy or chemoradiotherapy with one infusion of TILs when possible selected for E6 and E7 reactivity. Three patients experienced responses, with two CR and one PR; the two CR were ongoing 22 and 15 months after treatment [56].

Study C-145-04 [57] is an ongoing, open-label, multicenter phase 2 evaluating the safety and efficacy of LN-145 in patients with advanced CC in progression after 1st line of CT. Twenty-one patients have been treated so far, with an ORR of 44.4%, of which 11.1% are CR. The AE profile was generally consistent with the underlying advanced disease and in line with lymphodepletion & IL 2 regimens. These findings granted FDA’s Breakthrough Therapy designation on 22 May 2019.

TILs clinical application in CC is still at its initial stages, but the results are promising; especially in virus induced cancer, such as CC, the selection of viral antigen specific TILs, such as HPV E6/E7 can achieve a highly selective tumoricidal effect with fewer adverse events. Combination therapies (e.g., TILs + ICI) are worthy of further studies as well, with the possibility of a synergistic effect and a prolongation of the clinical benefit [58].

## 6. Antibody-Drug Conjugate (ADC): Tisotumab Vedotin

Tissue Factor (TF) is a fundamental part of the coagulation pathways in physiologic settings, but recent data has observed how the oncogenic environment can cause both constitutive and hypoxia-induced upregulation of TF in inflammatory, stromal and tumoral cells, aiding in angiogenesis, progression and metastases; furthermore, it is one of the links to the hemostatic system that contribute to a pro-thrombotic tumoral microenvironment [59]. High expression of TF has been found in several solid tumors, including CC, correlating to poor prognosis; its distribution and high speed of internalization make it a promising target for antibody drug conjugates (ADC).

Tisotumab-Vedotin (Figure 3) is a first-in-class ADC, comprising a tissue factor (TF)-specific, fully human monoclonal antibody conjugated to a microtubule-disrupting agent monomethyl auristatin E (MMAE) using a protease-cleavable linker. Tisotumab-Vedotin has shown activity both in vitro and mouse xenografts, by delivering MMAE to TF positive cells and inducing both direct cytotoxicity, as well as bystander killing, with minimal disruption of the coagulation cascade. Moreover, the compound was also able to stimulate immunogenic killing via Fcγ-mediated antibody-dependent cellular cytotoxicity (ADCC) and antibody-dependent cellular phagocytosis (ADCP) [60].

After innovaTV 201, the single arm phase I/II trial showed acceptable toxicities and promising activity in solid tumors that had progressed after at least two treatment lines [61], innovaTV 204 [62] enrolled 102 patients, confirming the results of the previous trial with an ORR of 24% (95% CI 16–33) with 7% CR and a disease control rate of 72%. The most common adverse events observed were alopecia, epistaxis, nausea, fatigue, myalgia and ocular symptoms, with a total of 92% of patients experiencing AE, but only 28% G ≥ 3.

The phase III innovaTV 301 trial is currently actively enrolling and features metastatic cervical cancer in the second or third line, randomized to receive Tisotumab-Vedotin or CT by physician’s choice.

## 7. Vaccines

Despite their reliability for preventing high risk types of HPV infection, vaccines are not effective in clearing already established chronic infections.

Currently, the most common treatment for high grade pre-cancerous lesions CIN2 or CIN3 is surgical excision through cold knife conization or loop electrosurgical excision; there are several complications related to these procedures including pre-term births, low birth weight, and increased in the need for a cesarian section [63].

The immune system has an important role in HPV-driven carcinogenesis, which seems to differ depending on the stage of development; in the early stages of viral infection, different mechanisms are put in place in order to escape immunity.

HPV gene expression is present throughout the layers of keratinocytes, but only in the outer layer of differentiated cells, is replication active and viral protein expression high. This, together with a lack of viremic phase, seems to aid in escaping immune recognition [64].

Furthermore, several pathways actively suppress immune activation, thus creating an immunosuppressive microenvironment, via the downregulation of INF production and signaling through E6, E7 and HPV8 proteins. Interestingly, IFN regulatory factor 3 (IRF3) is not efficiently repressed and induces cell autonomous immunity [65], which suggests it might be a promising target for anti-viral immunotherapy against HPV persistent infection. Furthermore, oncoproteins target the p300/CBP-associated factor/nuclear factor (NF)-κB pathway decreasing pro-inflammatory cytokines, such as IL-1β, but not IL-1α, providing another interesting therapeutic target that warrants further research [66]. The low levels of pro-inflammatory signals reduce the number of activated APC cells, further depleted by the suppression of chemokines, such as CCL20 by viral oncoproteins [67].

While in low grade dysplastic lesions, the mechanisms outlined persist, as the dysplasia increases there is also an important increment in the infiltration of inflammatory cells in the stromal tissue. Particularly, CCL2 production in monocytes is stimulated, which in turn causes higher levels of MMP-9. MMP-9 promotes vasculogenesis, a critical step towards malignancy [68], further proven by how MMP-9 blockage strongly impairs carcinogenesis in murine models [69].

The increase of CCL2 in monocytes is connected to an upregulation of IL-6 [70] and M-CSF [71], both crucial factors in the creation of a tumorigenic chronic inflammation environment.

Furthermore, IL-6 produced by high grade cervical cancer cells regulates not only the increase in expression of MMP-9 but also the decrease of CCR7, normally expressed on immature dendritic cells to ensure their responsiveness to lymph nodes guiding chemokines. CD83+ mature dendritic cells are present in the tumor stroma, but the lack of CCR7 and the upregulation of MMP-9 block effective antigen transport.

M2 polarized macrophages are also abundant in the cervical cancer microenvironment, induced by IL-6 and prostaglandin E2 production by CC cells; in vitro studies show high levels of tumor promoting factors, such as PD-L1, VEGF and IL10 when colorectal EGFR+ cancer cells were exposed to IgG anti EGFR (cetuximab) in an environment rich in M2 polarized macrophages, therefore, impairing cetuximab antitumoral activity [72]. High expression of PD-1 on M2 polarized macrophages coupled with PD-L1 expression on CD8+ T cells indicates a role in weakening their cytotoxic activity [66].

Given this peculiar immune microenvironment and its changes from viral infection to precancerous lesions to invasive malignant tumor discussed above, therapeutic vaccines are being explored in preclinical and clinical trials as a tool in both premalignant and malignant lesions

HPV E6 and E7 represent ideal targets for therapeutic vaccines, being necessary both in the initiation and maintaining the process of malignant formation, while L1 and L2, are the main targets for prophylactic vaccines, they are not expressed on premalignant and malignant cells; therefore, they are not viable once the HPV’s episome has integrated with the host genome.

Genetic (DNA, RNA, viral, bacterial), protein-based, peptide-based, or cell-based vaccines are currently being investigated.

In cell-based vaccines, dendritic cells (DC) are considered optimal candidates because of their role in initiating and expanding T cell response; autologous DC peptide pulsed with HPV-16 and HPV-18 E7 showed, in multiple studies, good immunological responses but lack of clinical activity, even when associated with stimulatory cytokines [73,74,75].

The acceptable safety profile and the proven immunological response certainly make DC based vaccines worthy of further investigation in order to improve clinical activity, although there are intrinsic limitations, such as being labor intensive and unlikely to achieve long term immunity because of the short half-life.

Peptide-based vaccines can utilize either short or long-chain peptides, with the latter being preferable due to not requiring MCH profile selection. As they tend to have low immunogenicity, they are often tested with immunogenic adjuvants.

A vaccine containing nine HPV-16 E6 and four HPV-16 E7 synthetic peptides reported 47% of CR at 12 months in patients affected by HPV-16 positive high-grade vulval intraepithelial neoplasia (VIN-3) [76].

Concerning vaccines based on bacterial vectors, *Lysteria Monocytogenes* has been utilized because of its ability to penetrate APCs and stimulate both innate and adaptive immunity. A phase II trial has observed an OS of 38% in CC patients in progression after one or two lines of CT treated with recombinant live attenuated *L. Monocytogenes* equipped in vitro with the ability to secrete the fusion protein E7- non-hemolytic listeriolysin O (Lm-LLO-E7). The phase III is currently ongoing and is expected to be completed in October 2024 [77].

In a phase I/IIa study, *Lactobacillus casei* expressing E7 (GLBL101c) was administered to patients diagnosed with CIN3; 30% showed regression to CIN1, 70% to CIN2 [78].

Viral vectors including adenoviruses, alphaviruses, and vaccinia viruses, are utilized to produce E2, E6 and E7, thus stimulating T cell immune responses. A vaccine comprised of modified vaccinia Ankara virus (MVA) as a vector for bovine HPV E2 [79] showed interesting activity in a phase III trial; patients with intraepithelial lesions were treated with local injections of MVA-E2, obtaining complete responses in 89.3% of female patients and 100% of men; 83% of patients did not have detectable HPV circulating DNA after the treatment [80].

Other viral vaccines are currently under study; TA-HPV (recombinant Vaccinia virus expressing E6 and E7) was characterized by an acceptable safety profile and immunogenicity in a phase II study [81], TG4001 (recombinant vaccinia virus Ankara containing the sequence coding for E6 and E7 and IL-2) was administered to CIN patients, obtaining mRNA clearance and CIN regression at colposcopy in 70% of cases [82].

DNA-based vaccines rely on the in-situ production of the antigen after the introduction in tissues of a DNA plasmid encoding for it, which results in the activation of helper T cells, antibodies and cytotoxic T cells. These vaccines do not produce neutralizing antibodies, allowing for repeated ministrations without the risk of a lower immune response [83].

Although early clinical data showed only moderate immunogenicity [84]. More recently, different strategies have been implemented to improve antigen processing, presentation and delivery. As of March 2021, the phase III trial REVEAL-1 evaluating the safety, tolerability and efficacy of VGX-3100 to treat HPV-16/18-associated cervical high-grade squamous intraepithelial lesions (HSIL) produced promising results, with a percentage of responders of 23.7% in the treatment group, versus 11.3% in the placebo group, with a safety profile consistent with earlier clinical trials [85].

Despite interesting results of all different forms of vaccines, none are currently approved in clinical practice; many have only been tested on small samples, safety concerns still need to be properly addressed, and efficacy, when compared to traditional surgical treatment for premalignant lesions, has not been addressed yet.

## 8. Discussion

Cervical cancer treatment has been addressed on multiple fronts in the past few decades; the introduction of a validated and easily reproducible screening method via the PAP smear has allowed early detection and a much better prognosis, while vaccines are an effective primary prevention tool. Both incidence and mortality have decreased in countries where the screening services are widely available, while anti HPV vaccines are predicted to have a great impact from the generation that had access to them onward.

Although great steps have already been taken, CC is still a clinical challenge; the PAP smear is not equally accessible in developing countries, patients might not be thorough enough with its timing, and false negatives are not insubstantial. Regarding the vaccine, the coverage is still suboptimal, and only the generations born after the early 90s have had access to it before having had sexual contact. Therefore, effective treatment options are still an unmet clinical need worthy of research, given the poor outcomes of patients diagnosed with metastatic or relapsed disease.

As discussed above, numerous important trials regarding immunotherapy have recently concluded successfully, with ICI being the most advanced, already approved after progression to the first line in PD-L1 positive patients. Although response rates remain on the lower end, dramatic and durable responses have been recorded, and given the limited options, in patients affected by metastatic or recurrent CC, ICI will certainly become an integral part of therapy, with the potential to be both a single agent and in combination with chemotherapy.

Any further investigation of new target therapies that show activity in preclinical settings should now be compared with the results obtained with the addition of ICI, given that it has proven to be more effective than the current standard of therapy in every setting.

Trials have been conducted, as stated above, in several clinical settings, from early disease to the later lines of systemic disease; this provokes a further line of questioning about the correct timing of immunotherapy. On one hand, offering the most effective option available as soon as permitted allows for a better chance at a durable or even complete remission of the disease; on the other hand, it would reduce even further the options should the cancer relapse. Further studies are also needed on the correct timing, association and sequencing and possible rechallenge of immunotherapy; it is widely established that chemotherapy promotes antitumor immunity through improving T cell infiltration and depleting the immunosuppressive populations. Drugs, such as anthracyclines, oxaliplatin, and paclitaxel are recognized to be inducers of immunogenic cell death [86].

Despite this, emerging data points towards a more complicated scenario; preclinical studies show that chemotherapy can also induce immunosuppressive cell populations [87].

Given the deep impact that chemotherapy and radiotherapy have on the tumor microenvironment and how crucial the TME is for the success or the failure of immunotherapy, correct timing and sequencing might greatly affect the responses and warrant further research and discussion.

The correct selection of patients through predictive biomarkers is another topic of essential importance.

PD-L1 expression, one of the most commonly chosen predictive markers, has been proven to be imperfect, because of the heterogeneity within the tumoral tissue, the possible variations as time progresses and the tumor is exposed to different therapies, and the variety of techniques used to assess it. Generally speaking, high PD-L1 expression is indicative of a good responder to ICI, but in multiple trials patients with low expression of PD-L1 have shown durable responses; therefore, treatment should not be restricted based solely on this factor.

PD-L2, a second less characterized ligand for PD-1, has been shown to be independently linked to responses to ICI in head and neck squamous cell carcinomas (HNSCC) [88]. In melanoma patients, CD8+ T cell density at the invasive margin has been proven to be a more reliable marker than PD-L1 and PD-1 expression.

In colon-rectal cancer the immunoscore, evaluating the density of CD3+, CD8+ T cells and CD45RO+ memory T cells better predicted the prognosis compared to TNM staging [89].

Furthermore, the limit after which the tumor is considered PD-L1 positive varies greatly both among different types of cancer and in different trials; a combined positive score (CPS) of 50% is necessary for NSCLC, of 10% for bladder cancer, all the way to only 1% for gastric and cervical cancer. The KEYNOTE 158 did not show any responses in PD-L1 negative disease; therefore, pembrolizumab was approved only for patients with a CPS >1%, but the interim analyses of KEYNOTE 826 show a benefit in PFS and OS regardless of PD-L1. The apparent contradiction between the trials highlights that PD-L1 is not a reliable predictive marker in CC and others.

In conclusion, immunotherapy has proven to be effective in CC, with advanced biotechnologies, such as TILs and ADC becoming very promising future options and ICIs confirming their potential in various clinical settings. Further corroboration with larger cohorts of patients is surely needed for the newer therapies, whereas more in-depth analyses of the correct timing, dosing, sequencing and selection of predictive biomarkers might impact the success rate of ICIs.

## Figures and Tables

**Figure 1 ijms-23-03559-f001:**
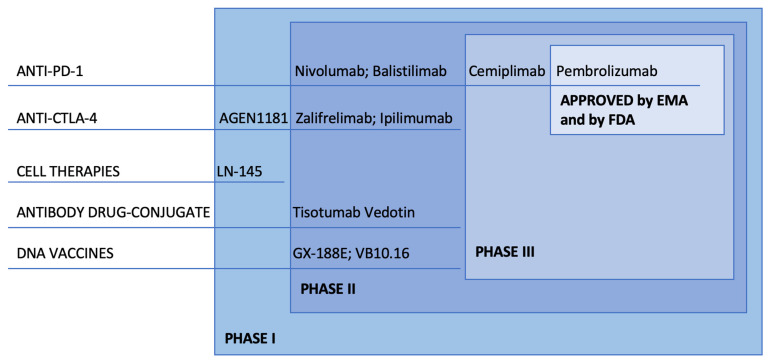
Most relevant immune therapies in development in Cervical Cancer.

**Figure 2 ijms-23-03559-f002:**
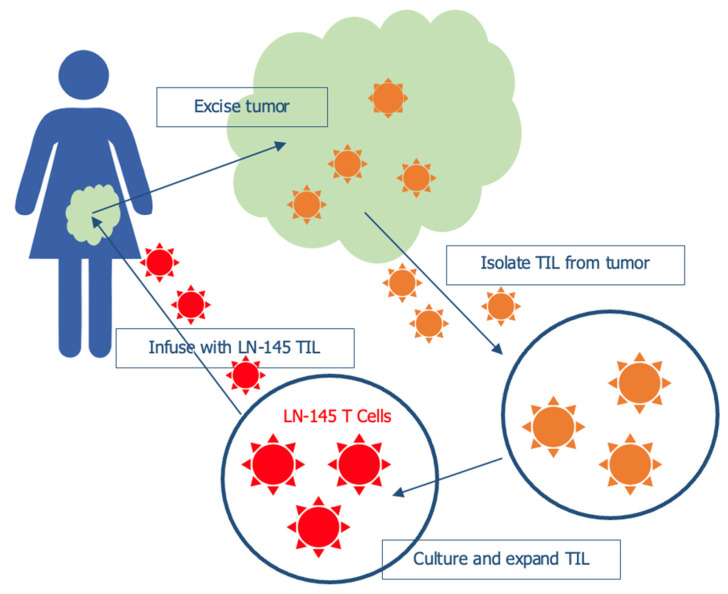
Tumor infiltrating lymphocytes (TILs) therapy.

**Figure 3 ijms-23-03559-f003:**
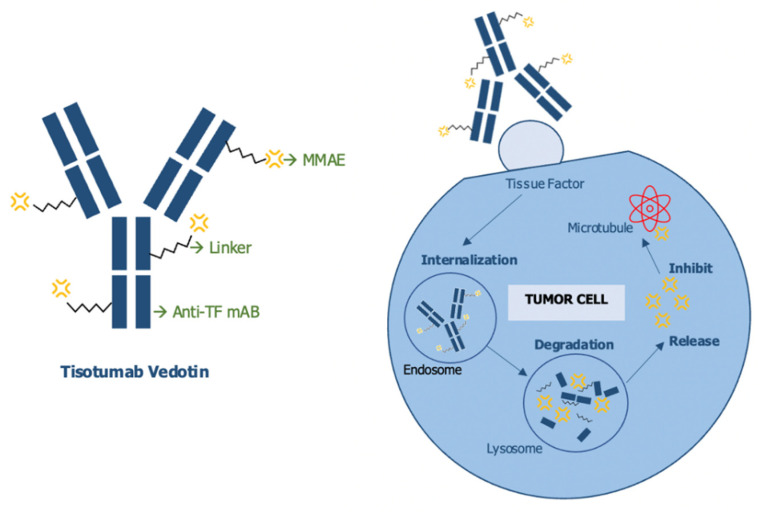
Tisotumab Vedotin molecular pathway of action.

**Table 1 ijms-23-03559-t001:** Most relevant ongoing clinical trials evaluating immune checkpoint inhibitors in the treatment of cervical cancer.

Trial	Phase	Number of Patients	Setting	Drugs and Schedule	Primary Endpoint
NCT03556839 (BEATcc)	III	404	Persistent Recurrent Metastatic	Arm A: Cisplatin 50 mg/m^2^ or carboplatin AUC 5 + paclitaxel 175 mg/m^2^ + bevacizumab 15 mg/kg q3W. Patients who achieve a CR after ≥6 cycles may be allowed to continue bevacizumab; Arm B: Cisplatin 50 mg/m^2^ or carboplatin AUC 5 + paclitaxel 175 mg/m^2^ + bevacizumab 15 mg/kg + atezolizumab 1200 mg q3W. Patients who achieve a CR after ≥6 cycles may be allowed to continue bevacizumab plus atezolizumab	OS
NCT04221945 (ENGOT-cx11/KEYNOTE-A18)	III	980	Locally advanced	Pembrolizumab 200 mg or placebo q3w for 5 cycles + CRT (weekly cisplatin 40 mg/m^2^ + external beam radiotherapy followed by brachytherapy) followed by 15 cycles of pembrolizumab 400 mg or placebo q6w	PFS and OS
NCT03830866 (CALLA)	III	770	Locally advanced	External beam radiotherapy with cisplatin (40 mg/m^2^) or carboplatin (AUC 2) once a week for 5 weeks, followed by brachytherapy, with durvalumab 1500 mg or placebo q4w for 24 cycles	PFS
NCT03104699	II	211	Locally advanced Recurrent Metastatic	Balstilimab 3 mg/kg q2w up to 2 years	ORR
NCT03495882	II	154	Locally advanced Recurrent Metastatic	Balstilimab 3 mg/kg q2w in combination with Zalifrelimab 1 mg/kg q6w up to 2 years	ORR
NCT03257267 (EMPOWER-GOG 3016/ENGOT-cx9)	III	608	Recurrent Metastatic	Experimental arm: Cemiplimab 350 mg intravenous administration every 3 weeks Investigator Choice Chemotherapy: - pemetrexed 500 mg/m^2^ q3w - topotecan 1 mg/m^2^ daily ×5 days, q3w - irinotecan 100 mg/m^2^ days 1, 8, 15, and 22, followed by 2 weeks rest, for 42 days (6-week cycle) - gemcitabine 1000 mg/m^2^ days 1 and 8, q3w - vinorelbine 30 mg/m^2^ days 1 and 8, q3w. Treatments will be given IV for up to 96 weeks	OS
NCT04238988 (CERV-3)	II	45	Locally advanced	Three cycles of NACT with carboplatin AUC 5, paclitaxel 175 mg/m^2^ and pembrolizumab 200 mg q3w, then surgery, then adjuvant carboplatin and paclitaxel in combination with pembrolizumab, followed by pembrolizumab 200 mg q3w for up to 35 cycles (only high risk)	2-years PFS
NCT03635567 (Keynote-826)	III	600	Persistent Recurrent Metastatic	Investigator Choice Chemotherapy: - paclitaxel 175 mg/m^2^ + cisplatin 50 mg/m^2^ - carboplatin AUC 5, with or Without bevacizumab 15 mg/kg) + pembrolizumab 200 mg or placebo q3w until disease progression, unacceptable toxicity or patient withdrawal for up to 35 cycles	PFS and OS
NCT04300647 (SKYSCRAPER-04)	II	172	Recurrent Metastatic	Atezolizumab 1200 mg q3w alone or in combination with tiragolumab 600 mg q3w	ORR

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
