# Peer review of "Immunotherapy for Cervical Cancer: Are We Ready for Prime Time?"

_ijms, 2022, doi:10.3390/ijms23073559_

Round 1

Reviewer 1 Report

Turinetto et al. have reviewed the published studies regarding therapeutic vaccines and immunotherapy in women with cervical cancer (CC). The rationale for employing immunotherapy in CC is strongly supported by multiple molecular features, such as high tumor mutational burden (TMB), microsatellite instability (MSI), high expression of programmed death ligand 1 (PD-L1) and high tumor inflammatory state. The only immune checkpoint inhibitors (ICI) currently approved by the Food and Drug Administration (FDA) is Pembrolizumab. Pembrolizumab is a highly selective, fully humanized monoclonal antibody that blocks the PD-1 receptor expressed on T cells and inhibits the PDL1 pathway. Radiation therapy can theoretically prime the immune system by releasing tumor neoantigens, thus creating an adaptive and cytotoxic T-cells mediated immune response that acts locally and distally. Implementation of Tumor infiltrating lymphocytes (TILs) has proven to be one of the most effective and specific adaptive cell therapies (ACT). HPV E6 and E7 represent ideal targets for therapeutic vaccines, being necessary both in the initiation and maintaining process of malignant formation. In conclusion, immunotherapy has proven to be effective in CC, with advanced biotechnologies such as TILs and ADC becoming very promising future options and ICIs confirming their potential in various clinical settings. Further corroboration with larger cohorts of patients is surely needed for the newer therapies, whereas a more in-depth analyses of the correct timing, dosing, sequencing and selection of predictive biomarkers might impact on the success rate of ICIs.

The claims are properly placed in the context of the previous literature. The experimental data support the claims. The manuscript is written clearly enough that most of it is understandable to non-specialists. The authors have provided adequate proof for their claims, without overselling them. The authors have treated the previous literature fairly. The paper offers enough details of methodology so that the experiments could be reproduced.

Minor revision

Line 477, "Conclusions" => "Discussion"

Author Response

Thanks for the valuable feedback. The change of the chapter title has been made.

Reviewer 2 Report

Well written, up-to-date summarized article about ICI & Cx cancer

Minor correction

1.Figure 1:Approved-->FDA? EMA?

2.Line 83:T cell activity and gen16erating -->16?

Author Response

Thanks for the feedback. Figure 1 has been changed to reflect both FDA and EMA’s approval, and the citation in line 83 has been corrected

Reviewer 3 Report

This is a timely and well put together review on an important topic that will be of interest to many researchers and clinicians. Some of the paragraphs are very short and this makes some sections feel disjointed.  These short sections could be looked at to improve the flow and readability. More on HPV and the role of HPV in cervical cancer is needed to help explain some sections. Note that immune responses to E2 were characterized many years ago as well as immune responses to other HPV proteins.

Specific comments

Line 28-36 - I would add more on current preventative vaccines and the role of HPV in cervical cancer - this would help later on when E6 and other viral proteins are mentioned but without introduction.

Line 78 - Human papilloma virus should be Human Papillomavirus   - this is now standard.

Line 83 - reference in the middle of a word

Line 80-86 - not clear if the authors are referring to PD-L1 expression on T cells or tumour cells - these could be clarified with a diagram.

Line 139 - innated should be innate

Line 178 passible ?   This sentence is not understandable as current written.  Passible is a archaic word and I don't think the authors mean to use it.

Line 206 Keytruda - name needs explaining as not explained or used earlier

Line 236 "About 46.2" is an odd way to day this

Line 258 in vivo is confusing in this context as this is in mice. Better to say in mouse models. More importantly the referenced paper did claim synergism BUT I don't think it demonstrated synergism. ie the effects look additive but not synergistic

Line 329 E6 and E7 come as a surprise here because the proteins were not mentioned earlier. It would be best to explain what they are earlier and explain what E6 and E7 reacting means

Line 348 thrombotic

Line 355 in vivo - might be best to specify mouse xenograft

Line 363 "with at least 2 lines solid tumours"   - what does this mean? It is confusing to me.

Line 404-408 sections needs referencing

Line 423 do the authors mean "chronically infected" or integrated HPV DNA with lost E1/E2 expression etc  This needs further explaining.

Line 452 there is no such thing as "bovine HPV E2". Also need to introduce E2 earlier

Author Response

Thanks for the valuable feedback. I have addressed below the changes that have been made after your comments.

Line 28-36 - I would add more on current preventative vaccines and the role of HPV in cervical cancer - this would help later on when E6 and other viral proteins are mentioned but without introduction.

We have expanded the topic to better introduce E6 and E7

Line 78 - Human papilloma virus should be Human Papillomavirus   - this is now standard.

We have corrected the term.

Line 83 - reference in the middle of a word

We have corrected the format.

Line 80-86 - not clear if the authors are referring to PD-L1 expression on T cells or tumour cells - these could be clarified with a diagram.

Both papers on the subject mention PD-L1 expression being high in tumor cells as well as mononuclear cells, I’ve corrected the sentence to be more explicit.

Line 139 - innated should be innate

We have corrected the grammar.

Line 178 passible ?   This sentence is not understandable as current written.  Passible is a archaic word and I don't think the authors mean to use it.

We have rephrased the concept in a more understandable format.

Line 206 Keytruda - name needs explaining as not explained or used earlier

We have added the chemical name.

Line 236 "About 46.2" is an odd way to day this

We have modified the phrasing.

Line 258 in vivo is confusing in this context as this is in mice. Better to say in mouse models. More importantly the referenced paper did claim synergism BUT I don't think it demonstrated synergism. ie the effects look additive but not synergistic

We have revised the paragraph to better reflect your comment.

Line 329 E6 and E7 come as a surprise here because the proteins were not mentioned earlier. It would be best to explain what they are earlier and explain what E6 and E7 reacting means

As addressed before we have more thoroughly expanded the topic

Line 348 thrombotic

We have fixed the grammar.

Line 355 in vivo - might be best to specify mouse xenograft

The term has been modified

Line 363 "with at least 2 lines solid tumours"   - what does this mean? It is confusing to me.

The sentence has been better phrased

Line 404-408 sections needs referencing

The appropriate reference has been added

Line 423 do the authors mean "chronically infected" or integrated HPV DNA with lost E1/E2 expression etc  This needs further explaining.

We have corrected the line

Line 452 there is no such thing as "bovine HPV E2". Also need to introduce E2 earlier

We have added another reference citing the use of bovine derived HPV E2, #79